# Single-Molecule Mechanics in Ligand Concentration Gradient

**DOI:** 10.3390/mi11020212

**Published:** 2020-02-19

**Authors:** Balázs Kretzer, Bálint Kiss, Hedvig Tordai, Gabriella Csík, Levente Herényi, Miklós Kellermayer

**Affiliations:** Department of Biophysics and Radiation Biology, Semmelweis University, 1094 Budapest, Hungary; kretzer.b1@gmail.com (B.K.); onlybalint@gmail.com (B.K.); tordaih@hegelab.org (H.T.); csik.gabriella@med.semmelweis-univ.hu (G.C.); herenyi.levente@med.semmelweis-univ.hu (L.H.)

**Keywords:** optical tweezers, concentration gradient, force spectroscopy, diffusion, microfluidics, fluorescence

## Abstract

Single-molecule experiments provide unique insights into the mechanisms of biomolecular phenomena. However, because varying the concentration of a solute usually requires the exchange of the entire solution around the molecule, ligand-concentration-dependent measurements on the same molecule pose a challenge. In the present work we exploited the fact that a diffusion-dependent concentration gradient arises in a laminar-flow microfluidic device, which may be utilized for controlling the concentration of the ligand that the mechanically manipulated single molecule is exposed to. We tested this experimental approach by exposing a λ-phage dsDNA molecule, held with a double-trap optical tweezers instrument, to diffusionally-controlled concentrations of SYTOX Orange (SxO) and tetrakis(4-N-methyl)pyridyl-porphyrin (TMPYP). We demonstrate that the experimental design allows access to transient-kinetic, equilibrium and ligand-concentration-dependent mechanical experiments on the very same single molecule.

## 1. Introduction

Single-molecule methods, which have been evolving progressively in the past thirty years [1,2,3,4,5,6,7,8,9,10,11], give unprecedented insights into mechanistic details of molecular phenomena because they provide the distribution of parameters beyond ensemble averages, reveal stochastic processes such as fluorescence blinking [12], uncover trajectories of processes that evolve along parallel pathways such as protein folding [13,14,15], and allow the characterization of mechanical functions and properties such as molecular elasticity and motor-enzyme force generation [16,17,18]. One of the major tools in single-molecule mechanics is optical tweezers [1,19,20,21], which have been successfully employed in the investigation of DNA [22,23], RNA [24] and proteins [13,15]. An important challenge in single-molecule investigations is to provide a suitable, possibly controllable, chemical environment. Optical tweezers may be combined with microfluidics to provide discrete changes in the solution conditions [25,26]. However, exposing a single molecule to continuously controlled concentration of solutes remains a challenge.

In the present work, we utilize the concentration gradient that evolves in a parallel, laminar-flow multichannel microfluidic device, to expose a model DNA molecule to controlled concentrations of model intercalators SYTOX Orange (SxO) and a porphyrin derivative (TMPYP). Such DNA-binding small molecules have long been of interest due to their widespread applications. They are often used as therapeutic drugs [27,28,29,30,31,32] or as building blocks in creating functional assemblies [33,34,35,36,37]. They are most frequently used in molecular biology as DNA-labeling agents that can also perturb enzymatic reactions [38]. They bind to DNA in several modes: major- and minor-groove binding, electrostatic/allosteric binding and intercalation [39]. A molecule may bind with multiple modes depending on the nucleotide sequence [40,41], and each mode may have different equilibration times [2]. The characteristic time for association and dissociation may vary extensively, ranging from less than a second to hours [38]. Thus, for a precise characterization of the chemical reactions under mechanical loads it is necessary to control the experimental parameters of force, ligand concentration and exposure time simultaneously and precisely. We show that by positioning a pre-stretched single λ-phage dsDNA molecule at different locations within a diffusionally controlled spatial concentration gradient of the DNA-binding ligand, the parameters of the binding reaction can be accessed by following the concentration-dependent changes in DNA extension. Furthermore, force-driven structural changes in the very same DNA molecule can be measured at well-controlled ligand concentrations. 

## 2. Materials and Methods

### 2.1. Sample Preparation

Biotinylated oligonucleotides (Sigma-Aldrich, St. Louis, MO, USA) were used to label the 3′ recessive ends of λ-phage dsDNA (Thermo Fisher Scientific, Waltham, MA, USA). Ligation was carried out with T7 DNA ligase (Thermo Fisher Scientific). Following biotin labeling the DNA samples were stored at 4 °C to prevent degradation caused by freezing [42]. For optical tweezer measurements DNA was used at 30 ng/mL concentration in Tris-NaCl buffer (20 mmol/L Tris-HCl, pH 7.4, 50 mmol/L NaCl) throughout the experiments. DNA molecules were tethered between 3.11 µm diameter streptavidin-coated polystyrene beads (Kisker Biotech, Steinfurt, Germany). As model ligands that bind DNA we used the fluorescent mono-intercalating dye SYTOX Orange (SxO) (Thermo Fisher Scientific) and tetrakis(4-N-methyl)pyridyl-porphyrin (TMPYP).

### 2.2. Single-Molecule Manipulation and Imaging

Single molecules of λ-phage DNA were mechanically manipulated by using a C-Trap dual trap optical tweezers system equipped with a multi-channel microfluidic flow-through sample chamber system and laser-scanning confocal fluorescence microscopy (Lumicks, Amsterdam, The Netherlands). The captured molecule was positioned at a distance ~100 µm away from the microchannel confluence to ensure a stable hydrodynamic environment. A typical solution flow rate of 1 mm/s was maintained throughout the experiments. Mechanical measurements were carried out in either constant stretch velocity (500 nm/s) or constant force (35 pN) modes. To measure fluorescence, a 532 nm YAG laser (Lumicks) was used for excitation. Fluorescence emission, across a wavelength range of 545–620 nm, was recorded with a photon-counting avalanche photodiode. We collected images either in the X–Y space, or kymograms in the molecular distance–time space.

### 2.3. Calibration of Concentration Gradient with Fluorescence

A diffusion-controlled ligand concentration gradient was generated between two, neighboring microfluidic channels, one of which contained high concentration of the DNA-binding molecule, and the other one containing buffer only. The shape of the concentration gradient depended on the limiting ligand concentrations (*c*_0_ = 0 and *c_s_* = 100 nmol/L), and the diffusion time (*t*) set by the flow velocity (*v* = 1000 µm/s) and the distance from the microchannel confluence (*d* ~ 100 µm) as *t* = *d/v* (see also Section 2.5 below) [43]. The actual spatial distribution of ligand concentration was characterized by imaging SxO fluorescence intensity along the gradient (i.e., in perpendicular to the flow direction) (Figure 1). The intensity profile (fluorescence intensity vs. distance) was calculated from the acquired image with the ImageJ software (version 1.52, public domain).

### 2.4. Molecular Scanning of the Ligand Concentration Gradient

Considering that the investigated ligands (SxO and TMPYP) intercalate into DNA and hence affect DNA’s contour length [38,40], the ligand concentration gradient may be mechanically mapped by moving a captured DNA along the gradient direction. We used two different methods: in the first, a single λ-phage dsDNA molecule, pulled taut with a constant force of 35 pN, was moved with constant velocity (18.5 µm/s) along the gradient (Figure 1). The molecular axis was parallel with the flow direction. Normalized length change was plotted against the distance traveled. In the second method, the dsDNA molecule was rapidly moved (with a speed of 500 µm/s) in discrete, 38 µm steps along the gradient. We plotted dsDNA length as a function of time. Whereas the first approach mimics an equilibrium experiment, the second approach is a transient kinetic one.

### 2.5. Theory

The evolution of the concentration gradient at the border between the neighboring microchannels is dictated by diffusion according to Fick’s second law, a one-dimensional solution of which is:
(1)$$c\left(x,t\right)=\frac{c_{s}}{2}\left[1+erf\left(\frac{x}{2\sqrt{Dt}}\right)\right]=\frac{c_{s}}{2}\left[1+erf\left(\frac{x}{\sqrt{2}\bar{x}}\right)\right],$$
where *c* is diffusant concentration, $c_{s}$ is the maximum, initial concentration in the dye channel, *D* is the diffusion coefficient, *t* is time, and *x* is distance along the one-dimensional diffusion coordinate (in perpendicular to the flow direction) so that *x* is 0 at the microchannel border and positive towards the dye channel. The parameter *erf* is an error function in the form of an integral of a Gaussian [44]. Considering that according to Einstein’s theory of diffusion the root-mean-square displacement of Brownian particle is $\bar{x}=\sqrt{2Dt}$, the term $2\sqrt{Dt}$ in Equation (1) may be replaced with $\sqrt{2}\bar{x}$ [45]. By removing *D* from the equation, we neglect the role of the diffusion coefficient, but we note that differences in the diffusibility of the different ligands alter the final shape of their respective concentration gradients. The boundary conditions of Equation (1) are such that the concentration gradients at infinite distances away from the microchannel border in either direction are 0. The initial conditions of Equation (1) are such that *c*(*x*,0) is 0 at x<0 and $c_{s}$ at x≥0 [44]. We used Equation (1) to fit the fluorescence intensity vs. distance profile across the microchannel border. 

## 3. Results and Discussion

In this work we investigated the applicability of a diffusionally generated concentration gradient in single-molecule biophysics. The concept of the experimental layout is shown in Figure 1. In the employed instrument force-measuring double-trap optical tweezers were combined with laser scanning confocal fluorescence imaging and laminar-flow microfluidics. Whereas the optical tweezers and fluorescence imaging allowed mechanical manipulation (molecule positioning, stretching, relaxing) and imaging, respectively, the multichannel microfluidic device enabled efficient solution control and the generation of a ligand concentration gradient at the border of vicinal microchannels. In the present experiments we manipulated λ-phage dsDNA as a model molecule, and used SYTOX Orange [38] and a porphyrin derivative (TMPYP) [34,41] as model DNA-binding ligands. To develop the ligand concentration gradient, one of the two neighboring laminar-flow microchannels (“dye channel” in Figure 1) contained high concentration of the ligand, whereas its neighbor only buffer (“buffer channel”). Such concentration gradients have in the past been employed for controlling cell behavior [46,47,48]. We note that in order to develop a concentration gradient in which the manipulated molecule responds across the entire concentration scale, the ligand concentration in the dye channel should be below the saturating concentration. Here, we used ligand concentrations of 100 nmol/L in the dye channel, which are close to saturation levels [38,40,41]. The concentration gradient develops as a function of time as the neighboring solutions flow, in parallel, with pre-adjusted velocity, down the microfluidic device. The shape (i.e., steepness) of the gradient can thus be chosen either by adjusting the flow velocity and/or by selecting the sampling position downstream of the microchannel confluence. We used a flow velocity of 1 mm/s throughout our experiments, and the sample (i.e., the trapped DNA molecule) was positioned at about 100 µm distal of the confluence.

We characterized the ligand concentration gradient by imaging SxO fluorescence and by mechanically scanning the gradient with the captured DNA molecule (Figure 2). Fluorescence imaging of the sampled microchannel border region (Figure 2A) revealed an area in which fluorescence intensity gradually increased from the buffer channel towards the dye channel. The corresponding intensity profile plot (Figure 2B) displayed a sigmoidal function which could be well fitted with Equation (1). The fit yielded 23.3 ± 0.1 µm for diffusion length, indicating that at the employed settings (i.e., maximum ligand concentration, flow velocity, distance from confluence) approximately 46 µm distance is available across which ligand concentration varies essentially linearly between 20–80 nmol/L. Mechanical scanning of the concentration gradient gave slightly different results (Figure 2C). It is possible to mechanically scan the ligand concentration gradients with dsDNA, because the binding of intercalating molecules results in DNA lengthening [38]. Accordingly, the DNA lengthening (normalized length increment) vs. distance (along concentration gradient) function displayed sigmoid curves for both SxO and TMPYP, although their trajectories were slightly different from that of fluorescence. In these experiments the DNA molecule was first pulled taut with a force of 35 pN (kept constant using feedback) so that its end-to-end length approximated its contour length, and the scanning of the gradient was carried out with a slow speed of 18.5 µm/s to approach chemical equilibrium. Fitting the SxO data with Equation (1) yielded 29.0 ± 0.2 µm for diffusion length, indicating that based on mechanical scanning about 58 µm distance is available across which ligand concentration varies essentially linearly between 20–80 nmol/L. Thus, mechanical scanning provides a more sensitive method of mapping the concentration gradient than fluorescence imaging with the current instrumentation settings. For practical purposes, an initial mechanical scanning of the concentration gradient with a DNA molecule is recommended, so that subsequent single-molecule mechanics experiments can be carried out at the desired ligand concentration. Altogether, these experiments provide the equilibrium length of the DNA molecule at the specific ligand concentration and at the specific, pre-adjusted force. The different shape of the extension-concentration trace observed in the case of TMPYP points at different binding mechanisms of TMPYP to DNA, which warrants further, detailed experimental exploration.

The use of SxO, a fluorescent intercalating dye, allows the direct imaging of the dsDNA molecule as it is translated across the concentration gradient (Figure 2D). The stark contrast between the stained DNA molecule and the microfluidic background emerges due to a ~450 fold increase of SxO fluorescence emission upon DNA binding [49]. As the captured DNA molecule was rapidly (with a speed of 500 µm/s) advanced to a new location along the SxO concentration gradient, its average fluorescence intensity increased and its contour length relaxed to an increased level (Figure 2E). Considering that at the employed speed the travel along the 38 µm step distance took only 76 ms, the subsequent lengthening, which relaxed fully in about five seconds (Figure 2F), corresponds to the kinetics of chemical equilibration rather than to the mechanical perturbation caused by the translational motion. Fitting single-exponential functions to the consecutive lengthening phases (Figure 2F) thus gave chemical equilibration time constants (*τ_eq_*) of 1.09, 1.22 and 1.23 s. In the reverse process, during which the DNA molecule was translated down the SxO gradient to the same spatial positions, we observed contraction relaxation steps (Figure 2F, inset). Fitting single-exponential functions to the consecutive contraction phases gave *τ_eq_* of 1.18, 1.18 and 1.68 s. Considering that in the last step the DNA molecule was positioned in an environment devoid of SxO, the equilibration time constant of this reaction corresponds to *τ_off_.* Notably, the obtained value (1.68 s) compares well to that measured earlier for the interaction of SxO with DNA at this force level [38]. Altogether, these experiments demonstrated that by combining optical tweezers with a pre-adjusted ligand concentration gradient a range of experiments can be carried out on the same single molecule, thereby providing rapid access to numerous equilibrium and kinetic parameters of a chemical reaction.

Our experimental setting allows the exploration of ligand concentration-dependent effects on the mechanical properties of a filamentous molecule. We investigated how increasing concentrations of SxO affect the force-extension curve of dsDNA by stretching and relaxing the same λ-phage DNA molecule at different spatial locations in the SxO gradient (Figure 3). Upon increasing SxO concentration the DNA molecule became longer (the non-linear force trace in the entropic regime shifted to greater lengths), and the cooperative overstretch transition and force hysteresis gradually disappeared in the explored extension range. The results show that the majority of the complex set of changes occurs below an SxO concentration of 15 nmol/L. By varying the parameters for generating the concentration gradient (flow velocity, ligand concentration in the dye channel, distance of sample location from microchannel confluence), mechanical measurements can be carried out across specific, narrow ligand concentration regimes, thereby uncovering further details of the molecular mechanisms of intercalator-DNA interactions.

## 4. Conclusions

In conclusion, in the present work we successfully employed diffusionally generated concentration gradients of DNA-binding ligands for measuring concentration-dependent effects on single-molecule mechanics. The combination of the double-trap optical tweezers with fluorescence imaging and carefully designed microfluidic environment allowed for rapid and efficient measurement of the mechanical properties of DNA exposed to different ligand concentrations of intercalators and provided access to numerous equilibrium and kinetic parameters of the ligand-DNA interaction. Our initial measurements on the comparison of SxO and TMPYP suggested that the binding of TMPYP to DNA is likely to follow mechanisms that are different from or additional to those of SxO. The major advantage of the use of the concentration gradient is that uncertainties of ligand concentration adjustment caused by the adsorption and desorption of cyanine dyes to and from the surfaces of the microfluidic device [38] can be alleviated. The concentration gradient may also be employed for investigating the effects of ligands (e.g., ions, substrates, etc.) on proteins. Although the small size of usual proteins may pose experimental challenge, large filamentous proteins (e.g., titin [50]) or proteins captured with DNA handles [51,52,53,54] may be investigated directly. Considering the emerging significance of single-molecule mechanics in understanding structural and functional detail, the addition of precisely adjusted ligand concentration gradients, as demonstrated here, may provide further access to understanding the exact mechanisms behind biomolecular phenomena.

## Figures and Tables

**Figure 1 micromachines-11-00212-f001:**
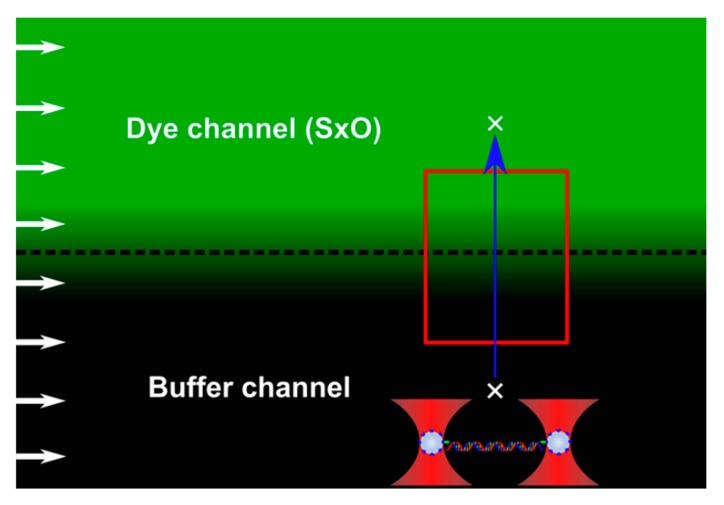
Schematics of the experimental design in the microfluidic system. White arrows on the left indicate constant flow velocity in the microchannels (dye and buffer channels). The dashed line marks the theoretical border between the two channels. Sytox Orange (SxO) concentration gradient arises due to diffusion between the neighboring microchannels, with the green background color corresponding to SxO fluorescence intensity hence the concentration. For the sake of simplicity, the progressive decay in the concentration gradient along the flow direction is neglected. The red rectangle indicates the area sampled either by the fluorescence or mechanical measurements. The blue arrow marks the path of the tethered DNA molecule, held by its ends with beads captured in independent optical traps (bottom of image), moved along the concentration gradient. White crosses mark the start and end positions of the mechanical trajectory.

**Figure 2 micromachines-11-00212-f002:**
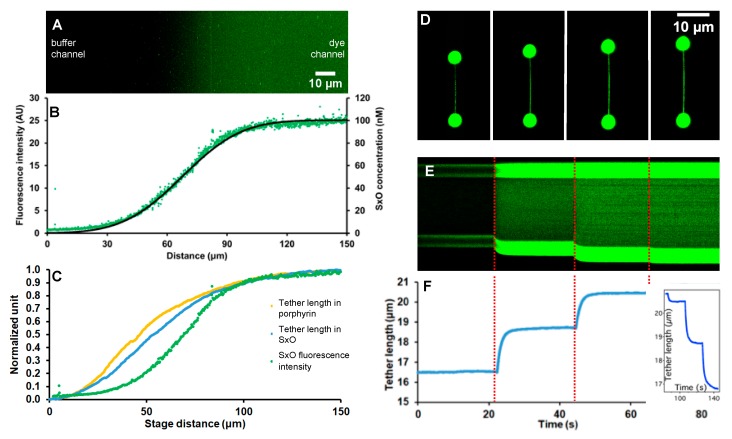
Characteristics of the ligand concentration gradient. (**A**) Laser scanning confocal microscopic image of the sampled microfluidic device area, across neighboring microchannels. The buffer and dye (SxO) channels are towards the left and right of the image, respectively. Initial SxO concentration (towards the right) is 100 nmol/L. Green coloring is artificial. (**B**) Fluorescence intensity (in arbitrary units, A.U.) measured along the SxO concentration gradient (green dots). Equation (1) was used to fit the experimental data (black continuous line). (**C**) Normalized DNA lengthening caused by mechanically sampling SxO (blue) and TMPYP (tetrakis(4-N-methyl)pyridyl-porphyrin, yellow) gradients by moving the DNA molecule, pulled taut with a constant 35 pN force, with a constant speed of 18.5 µm/s. For reference, the SxO fluorescence intensity data are also shown (green). (**D**) Images of DNA molecules, held stretched between two microbeads with constant force (35 pN), at four different positions (19, 57, 95, 133 µm, from left to right) along the SxO concentration gradient. The upper bead was re-positioned by the feedback system in order to maintain constant force. (**E**) Kymogram obtained by confocal scanning along the axis of the stretched DNA as a function of time during stepwise (38 µm/step) translation of the molecule along the SxO concentration gradient. Red dashed lines indicate the time points when the 38 μm steps were made. (**F**) DNA tether length as a function of time during rapid (500 µm/s), stepwise translation of the molecule along the SxO concentration gradient. In control experiments lacking SxO the thether length stayed constant. Inset shows the relaxation of tether length during stepwise translation of the same DNA molecule through the same spatial positions in the SxO concentration gradient. The non-linear decay curves, collected either towards or backwards the gradient, were fitted with the single-exponential function $L=L_{0}+\Delta Le^{-t/\tau_{eq}}$, where *L* and *L*_0_ are the actual and starting tether lengths, respectively, Δ*L* is the maximal tether-length change, *t* is time and *τ_eq_* is the equilibration time constant.

**Figure 3 micromachines-11-00212-f003:**
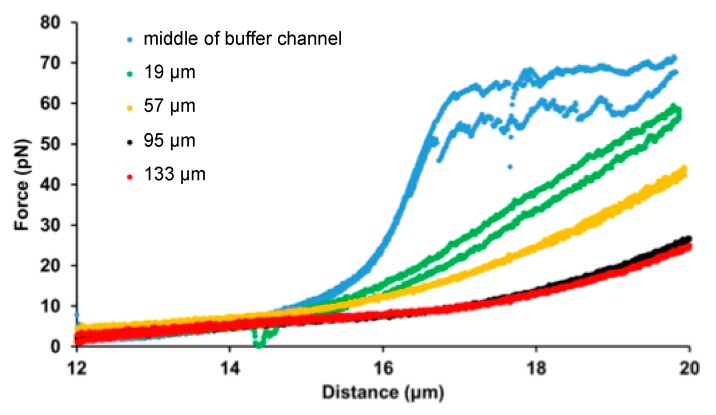
Constant-velocity stretch–relaxation cycles of DNA in a SxO concentration gradient. Force vs. extension curves for the same λ-phage dsDNA molecule were measured at five different locations of the microfluidic device (see legend): middle of the buffer channel (blue), and 19 (green), 57 (yellow), 95 (black), 133 (red) µm along the concentration gradient. The positions correspond to approximate SxO concentrations of 0, 15, 70, 95, and 98 nmol/L.

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
