# Peer review of "Single-Molecule Mechanics in Ligand Concentration Gradient"

_micromachines, 2020, doi:10.3390/mi11020212_

Round 1

Reviewer 1 Report

This is a very well written article and shows a nice - although not novel - application about how optical tweezers and microfluidic systems can be used to study single molecules. Lamda-phage dsDNA was optically trapped and moved with constant speed and force between two different solutions adjasent in a laminar flow where they slightly diffuse. SYTOX Orande or tretrakis(4-N-methyl)pyridyl-porphyrin were applied to flow laminarly along buffer solution. The strech of the DNA molecules was observed by confocal microscopy. The method is not new and is very well described in ref 26 of the manuscript. Although the experiment is neet and nicely performed, there is a lack of originality, novelty and/or purpose. Because of this a publication cannot be recommended.

Author Response

Please, see attachment.

Reviewer 2 Report

Single-molecule detection is a hot topic in molecular biology. The authors presented a smart way to measure concentration-dependent effects on single-molecule mechanics. The manuscript is well written, and the characterizations of the experiment are clear, therefore, I agree to accept this manuscript to be published on Micromachines

Author Response

We appreciate the enthusiasm and positive response of the reviewer about our manuscript.

Reviewer 3 Report

In the submitted manuscript, Kretzer et al developed a sophisticated method combining an optical tweezers instrument and a microfluidic system to enables analysis in ligand concentration gradients. This technical contribution to the field offers exciting promise for future uses. I list my comments below which would improve the manuscript before publication.

Specific comments:

Does the beads that translocate in the microfluidic system cause local disturbance of the ligand concentration? If the authors have data that show the ligand distribution around the beads with translocation, it would be better to add as a figure.

Fig. 2F, the inset is difficult to see. It should be revised.

Fig. 2F, Is the result shown in Fig. 2F can be due to DNA entanglement near the beads? Have the authors done the experiment without SxO (I mean, in the buffer only negative control)?

Is the system introduced in this work applicable to proteins instead of DNA? Please discuss about the possible application of the method.

Author Response

Please, see attachment.
